# Archaeomagnetic evidence indicates post-Inka reheating of metallurgical kilns at Quillay (NW Argentina)

Judit del Río[1], Miriam Gómez-Paccard[2]*, Alicia Palencia-Ortas[3], Annick Chauvin[4], Mara Basile[5], Norma Ratto[5], Pablo Cruz[6], Marco Antonio Giovannetti[7]

1 Departamento de Historia, Geografía y Comunicación, Universidad de Burgos, Burgos, Spain, 2 Instituto de Geociencias (IGEO), CSIC-UCM, Madrid, Spain, 3 E.T.S Ingeniería y Diseño Industrial, Universidad Politécnica de Madrid, Madrid, Spain, 4 Géosciences-Rennes - UMR 6118, Université Rennes – CNRS, Rennes, France, 5 Instituto de las Culturas, UBA – CONICET, Buenos Aires, Argentina, 6 Unidad Ejecutora en Ciencias Sociales Regionales y Humanidades, CONICET – Universidad Nacional de Jujuy, Jujuy, Argentina, 7 CONICET, Universidad Nacional de La Plata, La Plata, Argentina

* mgomezpaccard@csic.es

## Abstract

This study presents an archaeomagnetic analysis of five metallurgical kilns from the settlement of Quillay (Catamarca, NW Argentina), attributed to the Inka period on the basis of radiocarbon dates and archaeological evidence. All five structures yielded consistent paleofield values, and directional and intensity determinations are statistically indistinguishable which suggests that the sampled parts of the kilns were reheated within a relatively short interval. Comparison with geomagnetic field models, particularly BIGMUDI4k.1, indicates that this last heating event occurred between the late nineteenth and mid-twentieth centuries, rather than during the Inka occupation. The discrepancy between the archaeomagnetic and radiocarbon evidence is therefore consistent with a later remagnetization event affecting the structures. Possible explanations include a local wildfire or the deliberate re-use of the upper chambers to recover copper from slag. The results refine our knowledge about the occupational history of Quillay, and support the interpretation of a later thermal event affecting structures originally used in Inka times.

## Introduction

The geomagnetic field (GMF), generated by the movement of the convective flow of molten metal, primarily iron and nickel, in Earth's outer core, is one of the most striking characteristics of our planet. Before the advent of direct measurements in the 17th century, its past behavior can only be inferred from geological and archaeological materials that recorded the GMF's direction and intensity at the time of deposition or last heating. Expanding the archaeomagnetic database is therefore essential for reconstructing the configuration and temporal evolution of the GMF over historical

which permits unrestricted use, distribution, and reproduction in any medium, provided the original author and source are credited.

**Data availability statement:** The raw measurements data are hosted on Zenodo at https://doi.org/10.5281/zenodo.18341349. All other relevant data are within the man script and its Supporting information files.

**Funding:** This work was developed under the FPU20/03664 doctoral contract granted by the Spanish Ministry of Universities. The authors acknowledge support from the PID2020-113316GB-100, PID2024-159020NB-100 and PID2024-159094NB-I00 research projects, funded by the Spanish Ministry of Science, Innovation and Universities; the COOPB23002 cooperation project of the CSIC, Spain; and the PNP project funded by CNRS/INSU. We also extend our gratitude to the ArchaeologyHub. CSIC research network for its support. The funders had no role in study design, data collection and analysis, decision to publish, or preparation of the manuscript.

**Competing interests:** The authors declare no conflict of interest, whether financial or otherwise.

timescales, particularly in regions where data remain scarce, as is the case in most of the Southern Hemisphere.

Previous archaeomagnetic research in South America remains limited, representing only ~5% of the global database [1]. For the past two millennia, directional data are restricted to those from volcanic rocks in Chile [2,3] and a few archaeological sites in Bolivia and Peru [4]. Paleointensity datasets are comparatively more abundant [5–14], but most concentrate within the last millennium and often fail to meet current quality criteria [9–12]. This scarcity of reliable full-vector data from southern latitudes hampers robust reconstructions of past field behavior at both global and regional scales. To try to address this gap, we carried out an archaeomagnetic study in five metallurgical kilns at the site of Quillay (NW Argentina) dated to the Inka period (*ca.* 1400–1450 CE). Quillay emerges as an exceptional metallurgical center, the only site in the area with clear evidence of metalwork infrastructures, positioning it as one of the most important Inka metallurgical sites specialized in copperwork in the region.

## Archaeological setting

Quillay is located within the Pampas Sierras of Catamarca, at the confluence of the Hualfín River (also known as Belén) and its tributary, the Quillay River (27.43° S, 66.95º W, 1550 m.a.s.l; Fig 1). The valley hosted numerous pre-Hispanic settlements that played a key role in integrating northwestern Argentinian territories into the Tawantinsuyu, the Inka Empire. Among these sites are El Shincal de Quimivil, Los Colorados, Hualfín Inka, and Quillay itself [15–20].

Although many locations in the Southern Andes have yielded abundant evidence of advanced metallurgical practices [21–24], Quillay stands out as the only site in the Hualfín Valley, and one of the few of this age in the region, where purpose-built infrastructure for metalwork has been documented. Its exceptional concentration of furnaces, exceeding 30, further underscores its significance. The first archaeological interventions at Quillay date back to the 1950s, when González [25] reported an habitation area, an infant tomb, and pottery in various styles, alongside "a series of curious structures, shaped as truncated cones and of variable sizes" (25, p. 84, translation ours). A few decades later, the enigmatic structures were identified as metallurgical kilns [26]. Based on this evidence, the authors proposed a functional division of the settlement into two distinct sectors: one dedicated to domestic activities and another to metallurgical production, as indicated by the abundance of kilns. Recent research has confirmed this dual-purpose organization [27–30].

Five of these kilns were selected for this study and sampled during the Spring 2024 campaign. All share a similar morphology: they were built with bricks or baked clay and consist of two stacked pseudo-tubular chambers connected by perforated tunnels. Both chambers have circular or ellipsoidal bases, 100–130 cm in diameter, with walls approximately 10 cm thick. In addition, a rectangular antechamber precedes the lower chamber, serving as the entryway for fuel [28–30]. The full excavation of some of these kilns [28] revealed upper chambers with slightly concave walls narrowing towards the top, culminating in an opening that would have acted as a chimney. Fig 2 illustrates the architecture of the kilns.

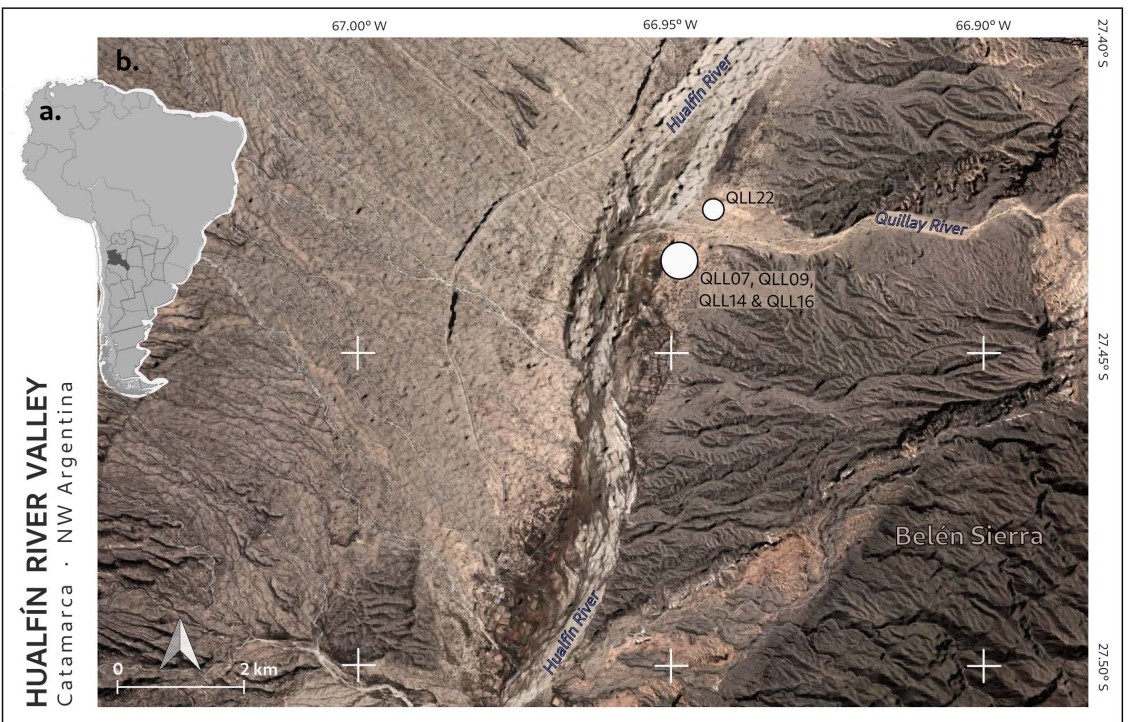

**Fig 1. Location of Quillay.** (A) Catamarca, where the site is located, highlighted on the South American map, (B) The central section of the Hualfín River Valley, withwhite dots indicating the kilns sampled in this study. Sources: Instituto Geográfico Nacional de la República Argentina and NASA ASTER V004.

Archaeomagnetic samples, indicated by the numbered white circles in Fig 2a, were collected from the exposed upper chamber, which was completely filled with sediments. As shown in Fig 2a (visible in the shaded area of the photograph) the inner wall exhibits evident signs of thermal alteration, visible as broad blackened zones. The lower chamber likely functioned as the firebox, where combustion of local woody species (e.g., *Prosopis sp.* or *Parkinsonia praecox*) occurred, possibly reaching temperatures up to 1000ºC [29]. This hypothesis is supported by the thick ash and charcoal layer found inside, as well as by the inspection of its inner surfaces, that revealed multiple metal slags, soot stains, and strong evidence of vitrification. In some kilns, accumulations of metal slags in the lower chamber suggest that it could have served to collect the melted alloy, too. Fig 2b shows a pseudo-cross-sectional reconstruction of kiln QLL22, based on [28,29].

## Radiocarbon dating

Three radiocarbon dates were previously available for kilns QLL09 [26] and QLL14 [28]; the latter obtained from charcoal of *Prosopis sp.* sourced from inka archaeological layers. Additional charcoal samples were collected from kilns QLL07, QLL16, and QLL22 during archaeomagnetic fieldwork, and sent to the Centro Nacional de Aceleradores in Sevilla (Spain) for dating. All radiocarbon results were calibrated using the SHCal20 curve [31] in OxCal 4.4 online [32] and are presented in Table 1.

At first glance, kiln QLL07 predates the other structures by nearly a century. After carefully reviewing and ruling out potential errors during the retrieval of charcoal samples, and considering that site-wide archaeological evidence does not reveal significant morphological or functional differences between this kiln and the others, we attribute this discrepancy to the old wood effect [33]. This well-documented phenomenon in radiocarbon arises from the longevity of certain species,

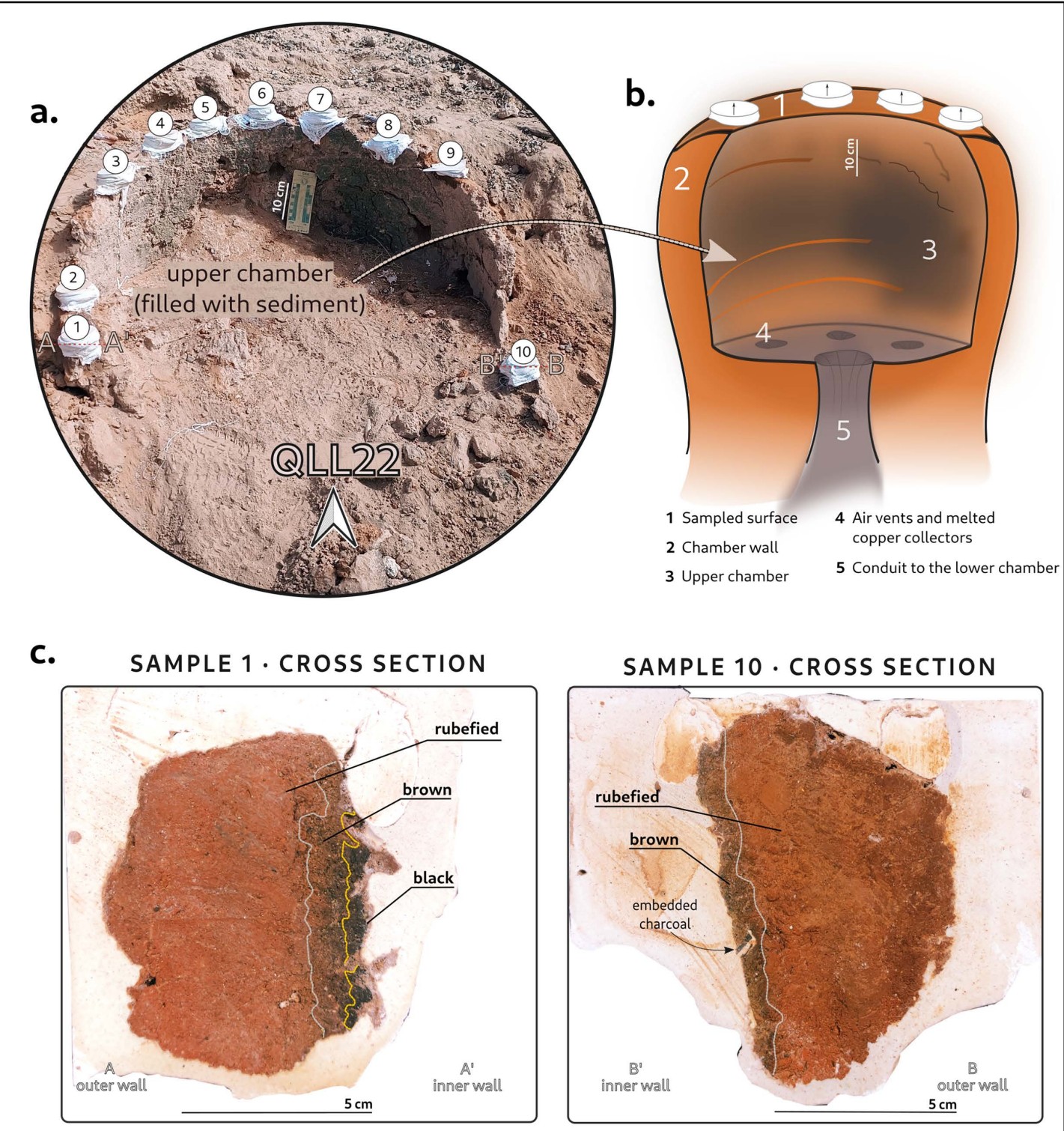

**Fig 2. Illustrations of kiln QLL22.** (A) The kiln during archaeomagnetic sampling; (B) Sketch reconstruction in pseudo-cross-section, depicting the upper chamber with vents on the floor and the tunnel that connected it with the lower chamber (not drawn); (C) Cross section of two archaeomagnetic samples photographed during laboratory sub-sampling prior to demagnetization. Lines indicate different facies: note the color variation with width, with darker zones corresponding to the innermost centimeters of the kiln's wall. Original image by the authors.

**Table 1. Radiocarbon dates for all sampled kilns at Quillay, expressed at 2σ. Samples marked with an asterisk were dated specifically for this study; the rest were retrieved from the available literature [26,28]. The δ13C (ratio of C12/C13) values signalled as "n/a" are not available.**

| KILN | MATERIAL | C14 SAMPLE | CONVENTIONAL RADIOCARBON AGE (YR BP) | δ13C (‰) | CALIBRATED RADIOCARBON AGE 2σ (CAL CE) | REFERENCE |
|---|---|---|---|---|---|---|
| QLL07 | wood charcoal | 7110.1.1* | 668±28 | −27.2 | 1296-1398 (95.4%) | This study |
| QLL09 | wood charcoal | AC-0553 | 390±100 | n/a | 1397-1696 (87.6%) | Raffino et al. (1996) |
| QLL14 | wood charcoal | LP-3198 | 560±50 | n/a | 1385-1458 (80.4%) | Spina (2019) |
| QLL14 | wood charcoal | LP-3216 | 510±50 | n/a | 1393-1504 (90.8%) | Spina (2019) |
| QLL16 | wood charcoal | 7111.1.1* | 511±28 | −22.5 | 1411-1457 (95.4%) | This study |
| QLL22 | wood charcoal | 7033.1.1* | 529±35 | −27.3 | 1401-1456 (95.4%) | This study |

which may be several decades or even centuries old when burned as fuel —likely the case of local carob trees (*Prosopis sp*.). Consequently, we do not consider this date as indicative of furnace's period of use. Note also the bigger uncertainties (larger intervals) shown by the earliest available dating (kiln QLL09), which reaches roughly three centuries [26].

Despite the aforementioned discrepancies, most C14 dates are consistent, and we deem those of kilns QLL14, QLL16, and QLL22 indicative of the whole ensemble. They place the archaeological layers of the kilns from which they were obtained between *ca.* the very end of the 14th century and the middle-to-end of the 15th century, agreeing with the archaeological evidence for the site based on ceramic wares and metallurgical production which postulates an active population at Quillay right before its integration into the Tawantinsuyu [17,27].

## Archaeomagnetic sampling and sub-sampling

The five kilns sampled exhibited varying degrees of erosion, from very deteriorated to nearly intact. Between 6 and 13 archaeomagnetic samples were collected per kiln, covering extensively the circumference of the extant walls that were visible above the ground. Preference was given to areas showing clear evidence of heat alteration, such as well-baked, reddened clay or blackened surfaces. No specific permits were required for this study, which complied with all relevant regulations for archaeological work in Argentina.

Sample extraction followed the plaster cap method (Fig 2a) with solar compass orientation. In the laboratory, samples were consolidated with sodium silicate before being embedded in a cubic mold filled with plaster. The resulting rectangular prisms, about 15x10x10cm, were then cut into slabs, which were consolidated a second time before being transformed into standardized cubic specimens of about 8 cm³. This sub-sampling protocol provides an opportunity to examine each hand sample at different stages of preparation. Fig 2c shows cross-section photographs of two samples from kiln QLL22. Differential thermal alteration is recognizable, with the innermost centimeters of the wall displaying darker tones ranging from deep black to dark brown. This visual evidence is particularly useful for identifying and selecting specimens that experienced the highest degree of thermal alteration.

## Experimental setup

Paleomagnetic and rock-magnetic experiments were performed at the Paleomagnetic laboratories of the University of Burgos and the University of Rennes. The natural remanent magnetization (NRM) was measured using 2G cryogenic magnetometers (2G Enterprises) and specimens were heated using a Thermal Demagnetizer MMTD80 (Magnetic

Measurements) and the locally-manufactured *Ramsés* furnace at Géosciences-Rennes. A total of 60 specimens (12 per kiln) were selected for full vector determination; additionally, 45 non-oriented specimens were used in intensity determinations. The classical Thellier protocol [34] was applied using a constant laboratory field of 45μT or 40μT, with temperature steps of 30 or 50ºC until complete demagnetization, typically around 520ºC. Partial thermoremanent magnetisation (pTRM) checks were performed every two temperature steps.

TRM anisotropy corrections, *abbr.* TRMani [35], were applied once specimens reached 60–70% demagnetization; cooling rate corrections, *abbr.* CR [36], were implemented right after the TRMani steps by extending the cooling time from the fan-assisted quick cooling time (1.5–3 hours) to *ca.* 24 hours to simulate natural cooling conditions. Both corrections were applied to every specimen. In addition, magnetic susceptibility ($\chi$) was measured after each temperature increase to monitor potential magneto-chemical changes induced by heating. Details on the experimental protocol can be found in S1 Appendix.

We adhered to strict quality criteria for archaeointensity determinations, which we consider essential for the robustness of data. Our data selection approach is consistent with recent high-quality archaeointensity and archaeodirectional studies, e.g., [37,38]. The measurements were uploaded to the MagIC database (earthref.org/MagIC/20562).

Given the visible color differences in the materials (Fig 2c), we defined three facies —*sensu* Goldberg & Macphail (2006): black (abbreviated *b* in specimen names), including vitrified, very hard and porous or "bubbly" areas, and a very thin greenish layer in the innermost millimeters; brown (abbr. *br*), typically a transitional area where the black facies progressively disappears; and rubified (abbr. *r*), light brown with red or orange undertones, characteristic of oxidative combustion processes. To characterize these facies, we performed a standard set of rock-magnetic experiments using a Variable Field Translation Balance (Magnetic Measurements). These included: (i) an IRM (isothermal remanent magnetization) acquisition curve up to 1T; (ii) a backfield curve (-1T); (iii) a hysteresis loop (±1T); (iv) and a thermo-magnetic curve with a pre-applied field of 1T, heating up to 600ºC and cooling back to room temperature. The data analysis was carried out in RockMagAnalyzer 1.1 [39]. Additionally, selected specimens were subjected to magnetic susceptibility vs. temperature measurements up to 700ºC using the Bartington MS3 X/T system at the Magma Laboratory of the Spanish National Geographic Institute (Fig B in S1 Appendix). A total of 95 bulk specimens representing all facies were analyzed.

## Results

### Rock magnetism

The facies-by-facies magneto-mineralogical analysis reveals consistent traits within each group. This variability is illustrated in Fig 3, which shows representative hysteresis loops and thermo-magnetic curves. All parameters derived from rock magnetism experiments can be found in Table C in S1 Appendix. Note that, for non-saturated samples, the hysteretic parameters may be underestimated on account of the impossibility of calculating the high-field slope portion of the loop [39]. The experiments are, nevertheless, useful in illustrating the magneto-mineralogical trend of the materials.

Black of vitrified specimens (Fig 3a) are remarkably similar across all five kilns. The hysteresis loops exhibit, generally, low coercivities and closed loops, with their coercive fields (Bc) ranging widely from 13 mT to 44 mT, and maximum saturation magnetization (Ms) between 1.4E-01 Am$^2$/kg and 9.9E-01 Am$^2$/kg. Their thermo-magnetic curves show a steady demagnetization trajectory with a distinct drop at 470–550ºC.

The brown facies specimens (Fig 3b) show very thin and fully closed hysteresis loops, similar to those of the black facies, with Bc ranging from 8 mT to 32 mT and and Ms between 1.9E-01 Am$^2$/kg and 8.8E-01 Am$^2$/kg. However, their Curie temperatures are higher, ranging from nearly 500ºC up to 580ºC.

Rubified specimens (Fig 3c and 3d) exhibit the greatest diversity of iron phases, including mixtures of low and high coercivity minerals, with Bc values between 10 mT and 25 mT and Ms between 1.5E-02 Am$^2$/kg and 6.5E-01 Am$^2$/kg. Some specimens (Fig 3c) contain predominantly low-coercivity minerals, characterized by tightly closed hysteresis branches, and thermo-magnetic curves with Curie temperatures close to 550–580ºC. Other specimens (Fig 3d), exhibit

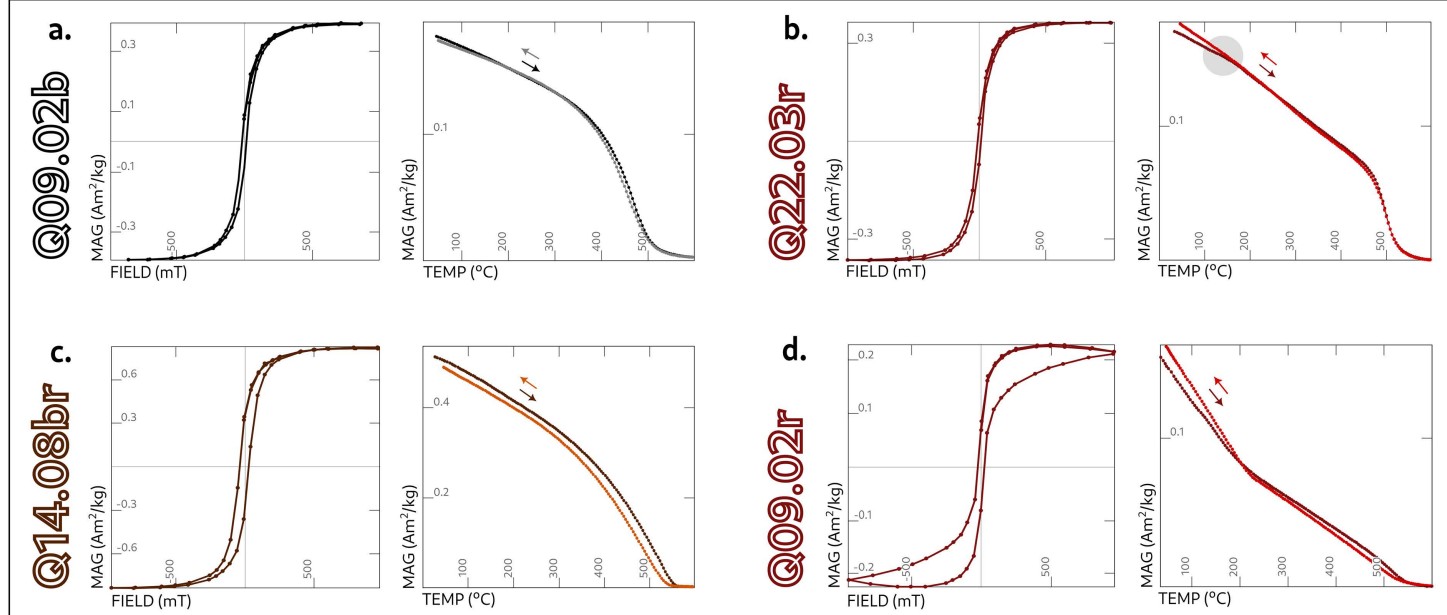

**Fig 3. Representative hysteresis loops corrected for para- and diamagnetic contributions (left) and thermo-magnetic curves (right) for different facies.** (A) Black facies from sample Q09.02. (B) Brown facies from sample Q14.08. (C) Rubified facies with low coercivity from sample Q22.03. The translucent gray circle on the thermo-magnetic curve highlights the increase and subsequent decrease in magnetization around 150ºC. (D) Rubified facies with high coercivity from sample Q09.02.

very high coercivity minerals with wide wasp-waisted hysteresis loops and almost linear thermo-magnetic curves, with a marked decrease in magnetization between 200–300ºC. They sometimes also display a peak between 100 and 200ºC (Fig 3c) that makes the cooling curve slightly irreversible from 200ºC downwards.

The low-coercivity magnetic carrier seems the same in all color facies, probably magnetite with a small percentage of cation substitution, either titanium or aluminum [40]. The high-coercivity fraction found in rubified specimens could correspond to an epsilon iron oxide [41–43], on account of the sudden decrease in the magnetization curves around 200–300ºC (Fig 3d). The bump around 100–200ºC (Fig 3c), in turn, could be due to the unblocking temperatures of single- or pseudo-single-domain, partially-substituted magnetite particles [44–46], or to the formation of new ε-Fe$_2$O$_3$ during the heating phase —perhaps a less-pronounced but similar behavior to the one we can observe in Fig 3d.

### Direction and intensity

Representative examples of the full-vector experiments are shown in Fig 4, including both accepted and rejected specimens.

For archaeointensity determinations, we adhered to up-to-date quality criteria for archaeological material following the guidelines in our previous work [47,48]. These include: (i) the linearity of NRM-TRM plots with no evidence of concavity suggesting multidomain (MD) behavior, (ii) at least 50% of the TRM used in slope calculation (f > 0.5), (iii) positive pTRM checks without evidence of mineralogical alterations, and (iv) single components of magnetization with MAD (maximum angle of deviation) and DANG (deviation angle) values lower than 5º.

As illustrated in the Arai plots in Fig 4a-e, the selected specimens fully meet these specifications. Rejected specimens either display erratic plots or exhibit double-component behavior, as evidenced in their Zijderveld plots (see example in Fig 4f). The application of the above-mentioned criteria resulted in a success rate of approximately 70%, with the oriented

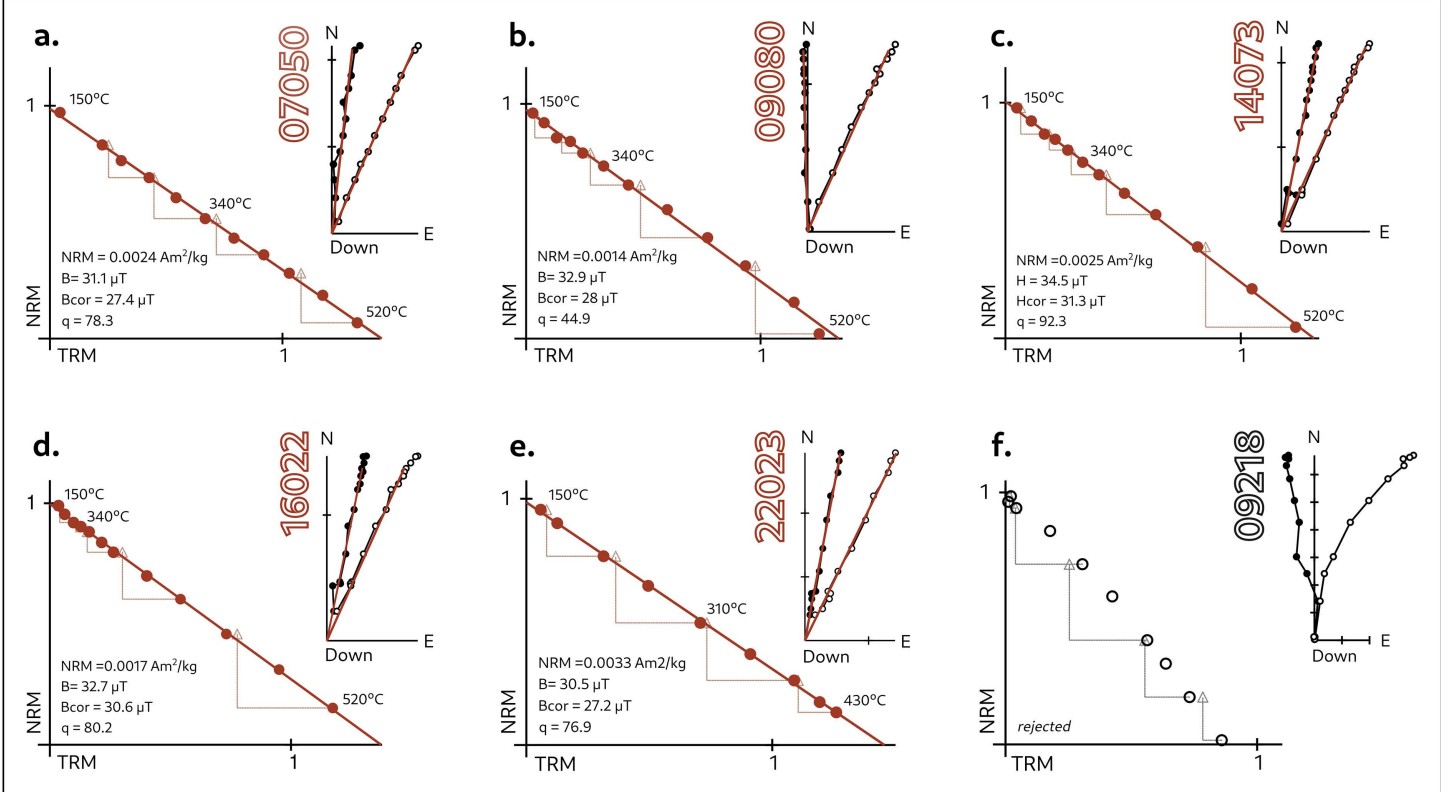

**Fig 4. Representative Arai diagrams and associated Zijderveld plots for accepted and rejected specimens.** For each specimen are provided: initial NRM, intensity estimates without corrections (B), intensity after TRM anisotropy and cooling rate corrections (Bcor), and quality parameter (q). On Zijderveld plots, black dots indicate the horizontal and white dots the vertical components. (A) kiln QLL07; (B) kiln QLL09; (C) kiln QLL14; (D) kiln QLL16; (E) kiln QLL22 (accepted specimens); and (F) kiln QLL09 (rejected specimen).

collection performing slightly better than the non-oriented one. Full results of the intensity determinations at the specimen level are provided in Table D in S1 Appendix. The mean intensities are provided as a weighted mean following the approach of Prévot et al.[49]. Comparison between uncorrected and fully corrected values (accounting for both TRMani and CR effects) revealed small differences, with corrected intensities averaging less than 2 µT lower. This confirms that neither TRM anisotropy nor CR effects significantly influenced the intensity record of these kilns. Nevertheless, we accepted the TRMani and CR corrected intensity values as final.

The mean direction recorded by each kiln was calculated from specimens previously accepted for archaeointensity determinations, using the same starting and ending temperatures. As shown in Fig 4, Zijderveld diagrams exhibit highly linear trajectories towards the origin, with almost complete demagnetization between 460ºC and 520ºC. Kiln-level averages were obtained following a hierarchical approach: first, sample means were obtained from their corresponding specimens; then furnace-level means were obtained from these intermediate sample averages. Directional means were calculated using the vector sum, and we assumed a Fisherian distribution [50] for the statistical parameters, as is the standard practice in archaeomagnetism. As is the case with intensity estimates, the effects of TRMani are not significant in directional determination; we nevertheless used the corrected directional values as final to remain consistent in our approach. The full results of the directional calculations at the specimen and sample level are provided in Tables E and F in S1 Appendix. The full-vector results for all five kilns are reported in Table 2 below.

**Table 2. Final full-vector results for all five kilns.** nd(ND): number of specimens (number of hand samples) used to calculate the directional mean; DEC: declination; INC: inclination; $\alpha_{95}$: semi-angle of 95% cone of confidence; k: precision parameter; ni(NI): number of specimens(number of hand samples) used to calculate the intensity mean; Bcor: intensity corrected for TRM anisotropy and cooling rate effects, with associated standard deviation.

| KILN | LAT (º) | LONG (º) | nd(ND) | DEC (º) | INC (º) | $\alpha_{95}$ (º) | k | ni(NI) | Bcor (µT) |
|---|---|---|---|---|---|---|---|---|---|
| QLL07 | −27.44 | −66.95 | 9(6) | 7.6 | −27.6 | 4.4 | 232 | 24(9) | 29.2±2.9 |
| QLL09 | −27.44 | −66.95 | 7(7) | 9.5 | −25.9 | 2.9 | 526 | 7(7) | 27.7±2.5 |
| QLL14 | −27.43 | −66.95 | 10(8) | 9.1 | −25.1 | 2.3 | 699 | 10(8) | 29.9±2.9 |
| QLL16 | −27.43 | −66.95 | 7(5) | 8.6 | −27 | 3.5 | 467 | 9(5) | 27.6±3.1 |
| QLL22 | −27.43 | −66.94 | 9(5) | 12.4 | −22.9 | 3.3 | 548 | 11(6) | 27.2±2 |

The five kilns yielded consistent archaeomagnetic results. Declinations (DEC) range from 7.6º to 12.4º, and inclinations (INC) from −22.9º to −27.6º, with their $\alpha_{95}$ values varying between 2.3º and 4.4º. Intensity values (Bcor) vary between 27.2µT and 29.9µT, with associated standard deviations from 2µT to 3.1µT. Both the directional and intensity results indicate minimal variability among kilns.

## Discussion

### Assessment of the full-vector values recorded by the structures at Quillay

Directional and intensity values from all kilns are consistent, with overlapping $\alpha_{95}$ confidence cones that suggest that the magnetic record of the structures is the result of a coeval use (Fig 5a). Importantly, the comparison of the GMF values with recent geomagnetic field models reveals a clear deviation between the empirical data obtained in this study and the expected field configuration at Quillay during the Inka period. This discrepancy is illustrated in Fig 5, where panels 5b–d compare our results with GMF parameters derived from three global geomagnetic models relocated to the site's coordinates: ArchKalmag14k.r [51], SHAWQ2k [52], and BIGMUDI4k.1 [53]. These models were selected because they rely only on paleomagnetic and archaeomagnetic data obtained from thermoremanent magnetization records, excluding sedimentary data that considerably smooth the models. SHAWQ2k extends to 1900 CE, ArchKalmag14k.r to 1950 CE, and BIG-MUDI4k.1 to 2000 CE; the latter also incorporates historical observations, available since the 15th century CE, providing the most detailed modelization for the contemporary era, which is of special interest for the present study.

When compared with the selected models, our data show declinations and inclinations that are systematically higher than the estimates for the Inka period, whereas intensity exhibits a pronounced low of approximately 30 µT. There are, thus, further questions that we need to address. First, what is the age of the last re-firing of the furnaces; second, what caused the fire that led to their remagnetization. We will discuss them next.

### Archaeomagnetic dating

Suspecting that the GMF record of Quillay's kilns is that of a rather modern time, we performed a test archaeomagnetic dating against the models discussed above [54]. Even though global models have inherent limitations discussed already in many works [55,56], since they do not include many datapoints for the Southern Hemisphere, they provide an overall view of the geomagnetic field that is useful in our study. Values similar to those recorded at Quillay are not expected in South America during the 15th Century, nor by global modelizations or by the existing regional intensity data from Argentina for the last 2000 years [6,9–11]. The results of the archaeomagnetic dating (Table 3) confirm that the last heating of the kilns did not occur during the Inka period, but rather during a reheating event in the contemporary era. Note that the early intervals at the beginning of the Common Era yielded by ArchKalMag indicate only that the configuration of the GMF

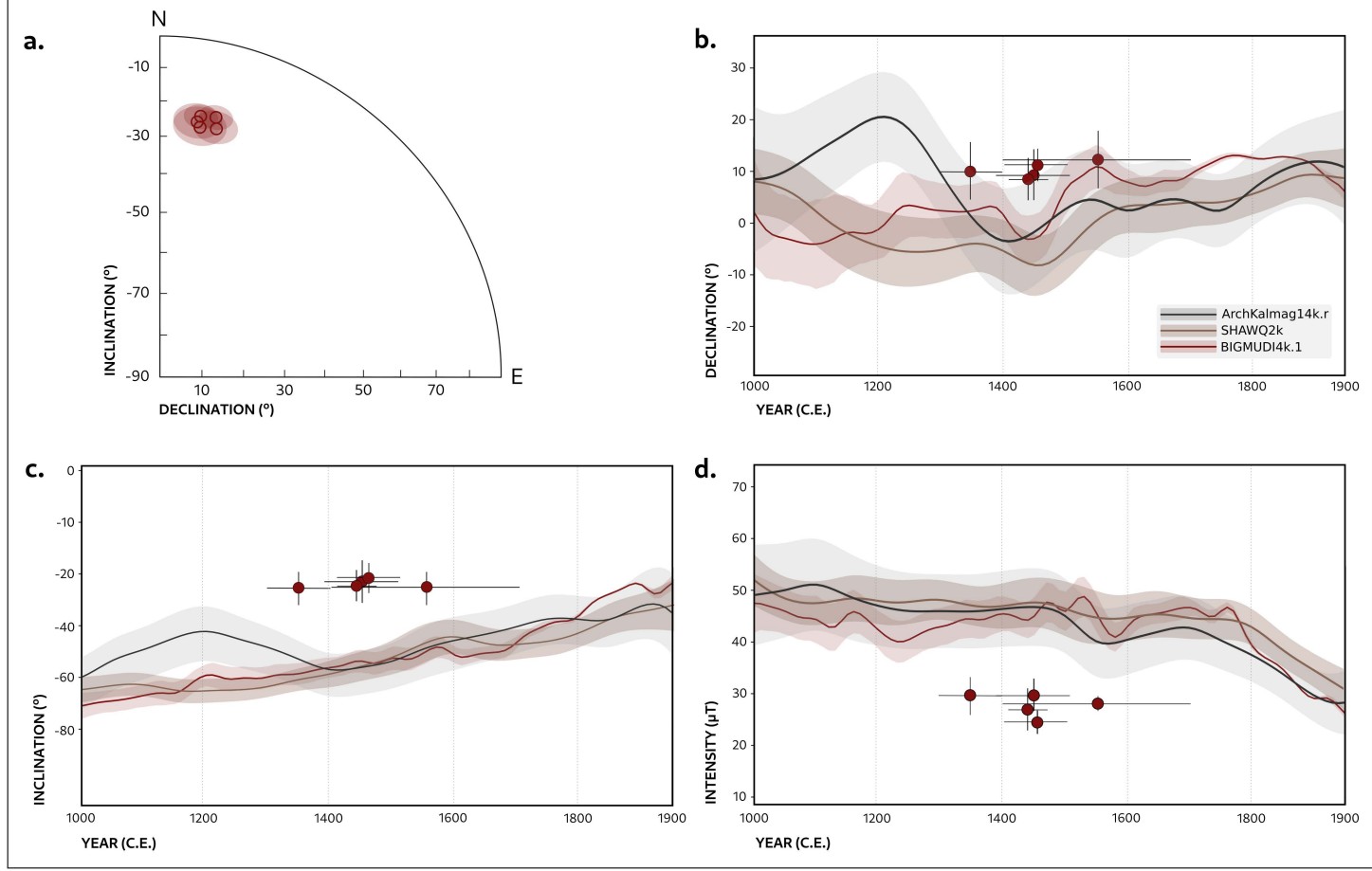

**Fig 5. Directional restuls for the five kilns, and full-vector comparison with global models.** (A) Directions recorded by the five kilns at Quillay; solid red dots represent mean directions per kiln, and translucent red areas indicate 95% confidence cones; (B-D). Declination, Inclination and Intensity at Quillay compared against the predictions of three recent global models: ArchKalmag14k.r (black), SHAWQ2k (brown), and BIGMUDI4k.1 (red). The datapoints in b, c, d are plotted assuming the average C14 dates reported in Table 1.

at that time was statistically compatible with those recorded by the kilns, but have been excluded due to archaeological and historical evidences in the Hualfín Valley. Two dating examples are illustrated in Fig 6.

The archaeomagnetic dating results indicate that the final use of all kilns occurred during the second half of the 19th century or later, irrespective of the reference model employed. It is important to note that the apparent upper limits provided by ArchKalmag14k.r and SHAWQ2k (1950 CE and 1900 CE, respectively) are constrained by the temporal extent of these models; consequently, these limits should not be interpreted as chronological boundaries. Among the models considered, BIGMUDI4k.1 offers the most robust estimates for this case study, as it integrates direct geomagnetic field measurements and extends to 2000 CE, thereby reducing truncation bias. A detailed assessment of the BIGMUDI4k.1 outputs indicates that all kilns were likely re-used between 1875 and 1945 CE, with some structures exhibiting narrower chronological ranges (e.g., QLL22: 1875–1905 CE) compared to others (e.g., QLL07: 1850–1950 CE), a difference that likely reflects the larger archaeomagnetic uncertainties (as per the $\alpha_{95}$ values) in the latter case.

However, as discussed earlier, radiocarbon and archaeological evidence unequivocally situates Quillay as an Inka settlement, which is demonstrated by the predominance of Belén-style ceramics dated to *ca.* 1480–1550 CE [19,57], their

**Table 3. Archaeomagnetic dating results obtained for the five kilns with the global models BIGMUD14K.1 [53], ArchKalMag14k.r(51), and SHAWQ2k(52) at 95% confidence. Results marked with an asterisk (\*) are included here for full transparency, but were rejected on the basis of archaeological evidence.**

| KILN | BIGMUD14K.1 | ARCHKALMAG14K.R | SHAWQ2K |
|---|---|---|---|
| QLL07 | 1900–1950 CE | 0–120 CE* | 1850–1900 CE |
|  |  | 1810–1950 CE |  |
| QLL09 | 1900–1945 CE | 1–70 CE* | 1865–1900 CE |
|  |  | 1845–1950 CE |  |
| QLL14 | 1900–1945 CE | 1–120 CE* | 1860–1900 CE |
|  |  | 1840–1950 CE |  |
| QLL16 | 1905–1945 CE | 1–95 CE* | 1855–1900 CE |
|  |  | 1835–1950 CE |  |
| QLL22 | 1875–1905 CE | 1860–1950 CE | 1875–1900 CE |

association with Inka wares [28], the presence of ceramic tools for characteristic metallurgical processes [58], and a textile fragment interpreted as part of a closure ritual in the lower chamber of kiln QLL14 [29]. The discrepancy between C14 and archaeomagnetic dates could be due to the provenance of the charcoals used in C14 dating, which were retrieved from the Inka stratigraphic units during the archaeological excavation of the kilns; note, however, that we have no information about the sourcing of the charcoals dated in earlier studies [26]. The radiocarbon dates (Table 1) would, then, correspond to the Inka use of the structures, while the archaeomagnetic dating (Table 3) marks the last heating-and-cooling event that the kilns sustained. Therefore, the archaeomagnetic results reveal a remagnetization process of the studied structures due to a re-heating occurred during the late 19th to early 20th centuries.

## Exploring the causes of contemporary re-magnetization

Given the morphological uniformity among all kilns and the archaeomagnetic evidence indicating very stable TRM values that are similar across all the studied structures, it is reasonable to infer that a localized, high-temperature heating event affected all discussed furnaces within a relatively short time frame. We present here two possible hypotheses that hold explanatory potential for such re-heating event: a generalized wildfire, and the intentional anthropic refiring of the structures.

Regarding the wildfire hypothesis, it is noteworthy that similar events have been documented at the neighboring archaeological site of La Ciénaga, located less than 10 km south of Quillay. Alosilla and colleagues [59] identified natural fires as one of the main factors causing severe damage to archaeological heritage in the Hualfín Valley, and local reports confirm that such episodes are recurrent in the region (59, p. 57). Based on the environmental and geomorphological similarity between both sites, we can assume that comparable events may have occurred at Quillay. Generally speaking, the environmental conditions in northwestern Argentina, and in Catamarca in particular, make the region highly susceptible to forest fires. This is consistently reflected in wildfire risk assessment reports issued by the Argentinian National Meteorological Service [60], which classify Catamarca as a high- to extreme-risk area during the summer months. The same concern is echoed in multiple reports by the Argentinian National Fire Prevention Agency over the years [61], underscoring the recurrent nature of these events and their potential impact on cultural heritage. Raffino and collaborators [26] already noted in their first field report about Quillay that the badland-like landscape with deep crevasses could have acted as a natural wind corridor that enhanced the efficiency of the smelting process; it is thus possible that this condition could also help the rapid propagation of wildfires, which would have found in the neighboring *Prosopis* forest a good source of fuel with high heat of combustion.

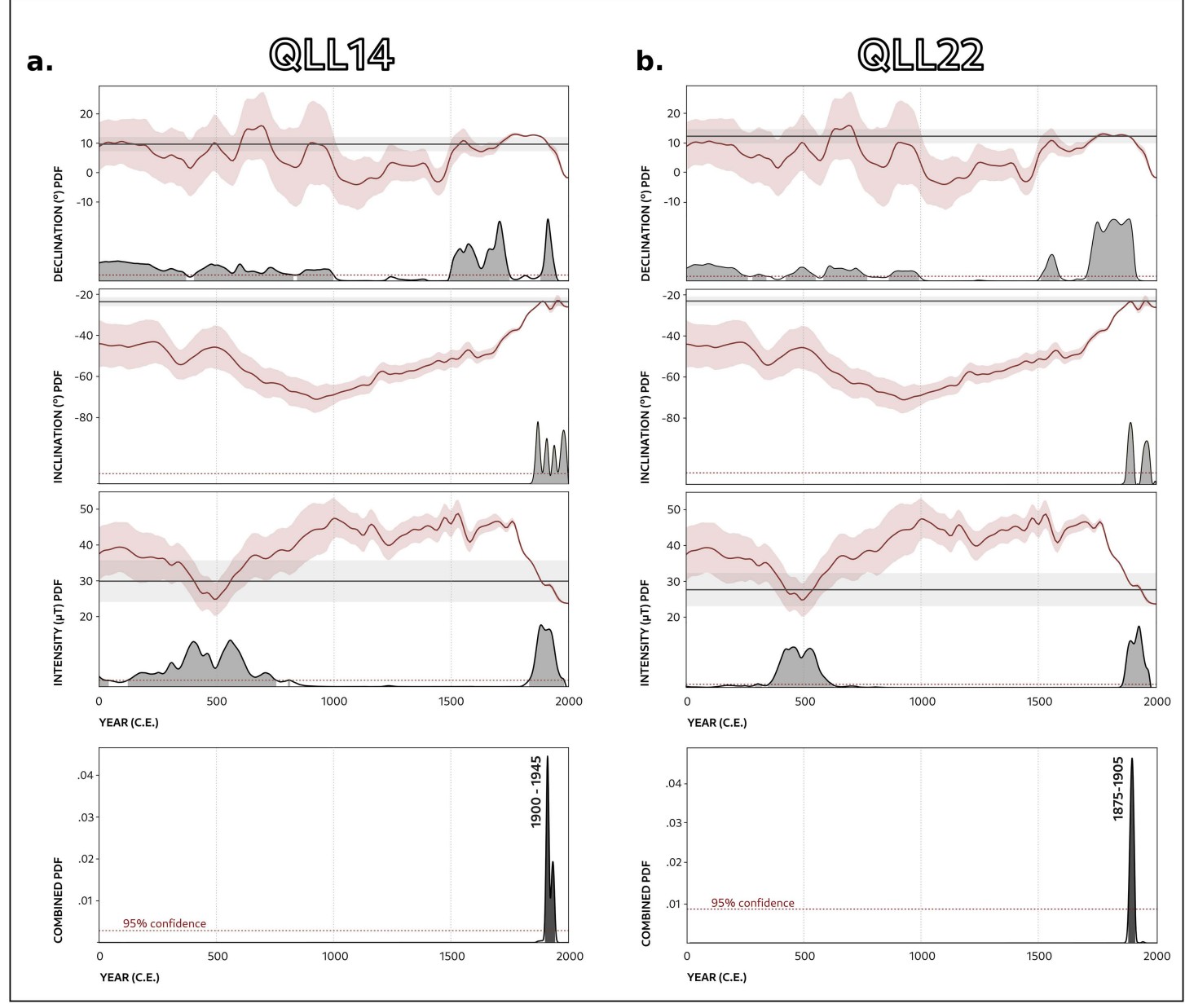

**Fig 6. Archaeomagnetic dating results for kilns QLL14 and QLL22 using the BIGMUDI4k.1 global model.** (A) Independent probability density functions (PDFs) for declination, inclination, and intensity; (B) Combined PDF integrating all three parameters and resulting age estimation for the kiln.

Another plausible explanation for the recent remagnetization of the kilns is the intentional re-firing of the furnaces by human agents, possibly to recover residual copper from the remaining slags through a secondary smelting process carried out directly within the upper chambers. Following a principle of economy, metalworkers would try to maximize the yields of their activity by reclaiming the copper still available from old facilities, before engaging in the more costly activity of ore selection, mining, and refining. This is well documented in both pre-Hispanic Andean metallurgical traditions and during the Spanish colonial period [62,63]; as well as in contemporary times [64–66], when these metal recycling practices

are locally known as *pirquineo*: precarious, small-scale metallurgical operations carried out by marginal or impoverished individual workers.

Traces of modern and contemporary *pirquinería* have been noticed in other archaeological studies of different chronologies in the Andes, too. Rato and collaborators [67], also working in Catamarca (Argentina), mention *pirquineo* as the main strategy for the exploitation of metallic ores in the region in the 20th Century. Van Buren and Mills [68] notice the continuous "harvesting" of metal slags at metallurgical facilities for the last 500 years in the Southern Andes. Their evidence is further strengthened by contemporary ethnographic observations at Porco (Bolivia): there, they witnessed a craftsman using a traditional wind-powered furnace (*wayra* or *huayrachina*) to smelt silver; moreover, they describe an almost-illicit trade of silver ore in which *pirquineros* collect small quantities of ore throughout the day to later sell it to these individual smelters. Godoy Orellana [64] reproduces an excerpt from an article appeared in 1825 on *El Eco de los Andes*, a briefly-lived local journal from the Argentinian province of Mendoza that explicitly mentions contemporary reuse of ancient metallurgical kilns. It reads: "Some elderly men acquired the habit of spending their Summers in the mines extracting metals, which they exploited in the absence of other silver miners, and which they smelted *in the many kilns that existed in the region from ancient times, whose remains can still be found*" (p. 46, translation and emphasis ours). These cases show the long-standing practices of reusing old facilities through time and reclaiming residual metal from them as a cost-effective alternative to the more labor-intensive processes of ore extraction and refining —of which Quillay may well be an example.

During a subsequent, shorter fieldwork campaign at the site, our team conducted preliminary X-Ray Fluorescence analyses on more than 70 samples in order to ascertain if any metallic elements could be found. The results were negative. This virtual absence of metallic residues or slag fragments in the upper chambers agrees with a similar observations by other authors in the Southern Andes [63,68]: Lechtman, for instance, observed the near absence of metal scraps, slags, or ores at Peruvian Colonial metallurgical sites, noting that "such slags have been claimed as mining 'sites' and have been removed and re-smelted for the metal they contained" [63, p. 31]. It was also during this second campaign that big accumulations of charcoal were found on the vicinity of kilns QLL14, QLL16, and QLL22 (Fig 7); their superficial position points to their recent origin, which could be due to the aforementioned copper recycling activities.

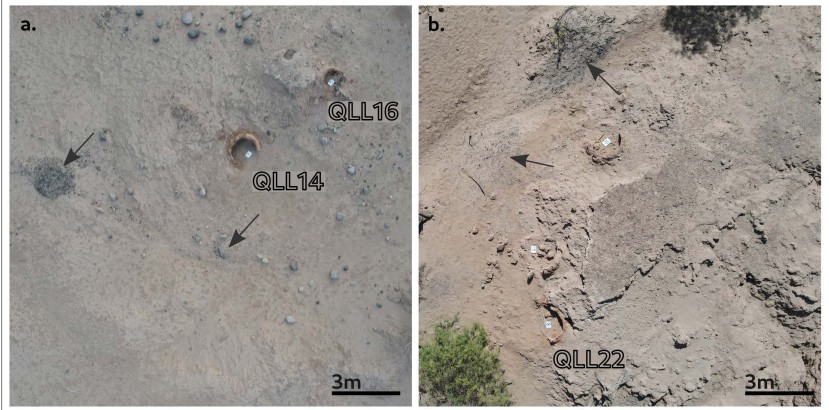

**Fig 7. Drone photographs.** (A) kilns QLL14, QLL16 and (B) QLL22. The black arrows mark superficial charcoal accumulations. Note that in (B) there are two more kilns that remained unsampled. Photographs by the authors.

## Limitations of the study and further work

The archaeomagnetic evidence presented here only allows us to assert the existence of a high-temperature heating event that affected all five sampled structures at Quillay, not its causes. Our current understanding of the site and the available environmental, archaeological, and ethnographical data (discussed in the previous sub-section) support both the wildfire and the intentional reuse hypotheses, which prevents us from favoring one over the other. Note that the campaign to sample the materials for this study was conducted only on the upper chambers due to the nature of archaeological work, which demands rigorous documentation of all structures that are about to disappear. This means that, in order to sample the lower, unexposed chambers, a complete archaeological excavation must be conducted, a lengthy process that requires time and resources currently not within our possibilities, and to which the remote location of Quillay adds another layer of difficulty. We hereby acknowledge the limitations of our study to provide definitive answers while raising fascinating questions, and the need of a future integrated multi-disciplinary study to decipher the origin of the contemporary remagnetization of the kilns.

In order to confirm the hypothesis of the occurrence of wildfires at Quillay, similar to those that have happened elsewhere in the Hualfín Valley, a broader landscape archaeology study would be needed. This should include the systematic surveying of adjacent areas to identify recent ash deposits and fire destruction layers that could subsequently be sampled and dated. An ancillary archaeomagnetic study of different kilns at Quillay could also be beneficial, since it would allow us to understand whether or not the re-heating episode was generalized at the site. At present, there is no evidence at Quillay indicating the occurrence of a major wildfire, such as extensive ash deposits, massive charcoal concentrations all over the site, or completely burnt structural remains which could be excavated; nor do historical records mention such an event.

To advance the re-smelting hypothesis, a new archaeomagnetic study should be conducted on samples retrieved from the lower chambers of the same kilns reported here, which are assumed not to have been reheated. By comparing the GMF values from both chambers, it would be possible to determine unequivocally whether the firing episode identified in this work occurred exclusively in the exposed upper chambers, or if it involved the entire kiln. The latter would have been a more demanding activity requiring the unearthing of the lower chamber and the removal of the accumulated sediment that fills it by the new users before they could carry out smelting activities —our assumption is that the simultaneous remagnetization of both chambers cannot be due to natural fires, during which the lower parts would have remained buried, as we find them today. In addition, new C14 datings from accurately provenanced materials should be carried out, both from different stratigraphic units in the set of kilns that comprise this study, and for the superficial charcoal accumulations showed in Fig 7. The assessment of these new dates will provide key information in the chronological characterization of the kilns' and site's use.

Ideally, conducting a survey on the Hualfín Valley at Quillay's coordinates in search of episodes of wildfire, and re-sampling the kilns here reported together with new ones are strategies that should be implemented together in order to substantiate any of the proposed hypotheses, which are currently tentative interpretations of our results. While all available archaeological evidence confirms that Quillay was an Inka settlement, the archaeomagnetic data provide new insights into the transformations experienced by the metallurgical kilns after their abandonment, while at the same time raising intriguing questions about the causes of the identified contemporary heating episode —questions that, for the time being, remain open.

## Concluding remarks

This study presents the archaeomagnetic analysis of five metallurgical kilns from the Inka settlement of Quillay in NW Argentina. The kilns exhibit statistically indistinguishable directional and intensity values, indicating that the remanent magnetization acquired by samples from their upper chambers was recorded at very similar ages. The comparison of the registered values with predictions from global geomagnetic field models at the site's coordinates shows that the

archaeomagnetic record does not match the expected values for the Inka period. Archaeomagnetic dating using the BIG-MUDI4k.1 global model, which incorporates direct observations, archaeomagnetic and volcanic data, indicates that the last firing of the upper chambers of these kilns occurred at the beginning of the 20th century or later. This shows a marked disagreement with previously available C14 dates, which place the kilns within the Inka period. We interpret this discrepancy as radiocarbon dates referring to the Inka use of the kilns, and archaeomagnetic dates referring to a later re-heating event. We propose two possible explanations for this contemporary reheating: (i) a large-scale wildfire affecting the site, or (ii) the intentional re-firing of the upper chambers to recycle copper minerals remaining in the slags. Further work is needed to ascertain any of these hypotheses.

## Supporting information

**S1 Appendix. The Supporting Information for this article is available as a separate file (S1 Appendix).** In addition, the experimental data can be found at doi.org/10.5281/zenodo.18341349 and earthref.org/MagIC/20562. (PDF)

## Acknowledgments

The authors thank Dr. A. Kosterov, and two more anonymous reviewers for their suggestions, who have improved the quality and accuracy of our manuscript. We extend our gratitude to our editor, whose care and comments have been important in the development of the finished article. Thanks also to the ArchaeologyHub.CSIC research network for its support. JdR has a debt of gratitude with Dr. Víctor Villasante, not only for allowing the use of the facilities at the L-MAGMA laboratory (Spanish National Geographic Institute) but for his guidance and kindness through the years. She also thanks the personnel at Géosciences Rennes for carrying out part of the labwork; Dr. Pierrick Roperch both for developing the Starmac suite and for his support during data analysis; and most importantly the people at Quillay.

## Author contributions

**Conceptualization:** Judit del Río, Miriam Gómez-Paccard, Alicia Palencia-Ortas, Mara Basile, Norma Ratto, Marco Antonio Giovannetti.

**Data curation:** Judit del Río.

**Formal analysis:** Judit del Río.

**Funding acquisition:** Miriam Gómez-Paccard.

**Investigation:** Judit del Río, Miriam Gómez-Paccard, Alicia Palencia-Ortas, Annick Chauvin, Mara Basile, Norma Ratto, Pablo Cruz.

**Project administration:** Miriam Gómez-Paccard.

**Resources:** Miriam Gómez-Paccard, Annick Chauvin, Norma Ratto.

**Supervision:** Miriam Gómez-Paccard, Alicia Palencia-Ortas.

**Validation:** Judit del Río, Miriam Gómez-Paccard, Alicia Palencia-Ortas, Annick Chauvin, Mara Basile, Norma Ratto, Pablo Cruz, Marco Antonio Giovannetti.

**Visualization:** Judit del Río.

**Writing – original draft:** Judit del Río.

**Writing – review & editing:** Judit del Río, Miriam Gómez-Paccard, Alicia Palencia-Ortas, Annick Chauvin, Mara Basile, Norma Ratto, Pablo Cruz, Marco Antonio Giovannetti.

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
