## [Decision Letter · Decision Letter 0]

5 Jan 2026

PONE-D-25-64919Slag re-smelting in modern times: archaeomagnetic evidence at the Inka site of Quillay (NW Argentina)PLOS One

Dear Dr. Gómez-Paccard,

Thank you for submitting your manuscript to PLOS ONE. After careful consideration, we feel that it has merit but does not fully meet PLOS ONE’s publication criteria as it currently stands. Therefore, we invite you to submit a revised version of the manuscript that addresses the points raised during the review process.

**The reviewers broadly agree that the study is based on careful field and laboratory work, but they differ in their assessment of the strength of the interpretative claims. In light of these reviews and my own evaluation, substantial revisions are required to bring the interpretation, structure, and presentation of the manuscript into closer alignment with the evidence presented. In particular, revisions addressing overstatement of conclusions, separation of results from interpretation, clarity of figures and tables, and consistency between the main text and Supporting Information are required for further consideration of the manuscript.**

We look forward to receiving your revised manuscript.

Kind regards,

Przemysław Mroczek, Dr. hab.

Academic Editor

PLOS One

**Journal Requirements:**

1. When submitting your revision, we need you to address these additional requirements. Please ensure that your manuscript meets PLOS ONE's style requirements, including those for file naming. The PLOS ONE style templates can be found at https://journals.plos.org/plosone/s/file?id=wjVg/PLOSOne_formatting_sample_main_body.pdf and https://journals.plos.org/plosone/s/file?id=ba62/PLOSOne_formatting_sample_title_authors_affiliations.pdf 2. Please include a complete copy of PLOS’ questionnaire on inclusivity in global research in your revised manuscript. Our policy for research in this area aims to improve transparency in the reporting of research performed outside of researchers’ own country or community. The policy applies to researchers who have travelled to a different country to conduct research, research with Indigenous populations or their lands, and research on cultural artefacts. The questionnaire can also be requested at the journal’s discretion for any other submissions, even if these conditions are not met.  Please find more information on the policy and a link to download a blank copy of the questionnaire here: https://journals.plos.org/plosone/s/best-practices-in-research-reporting. Please upload a completed version of your questionnaire as Supporting Information when you resubmit your manuscript. 3. In your manuscript, please provide additional information regarding the specimens used in your study. Ensure that you have reported human remain specimen numbers and complete repository information, including museum name and geographic location.  If permits were required, please ensure that you have provided details for all permits that were obtained, including the full name of the issuing authority, and add the following statement: 'All necessary permits were obtained for the described study, which complied with all relevant regulations.' If no permits were required, please include the following statement: 'No permits were required for the described study, which complied with all relevant regulations.' For more information on PLOS One's requirements for paleontology and archeology research, see https://journals.plos.org/plosone/s/submission-guidelines#loc-paleontology-and-archaeology-research. 4. We note that the grant information you provided in the ‘Funding Information’ and ‘Financial Disclosure’ sections do not match.  When you resubmit, please ensure that you provide the correct grant numbers for the awards you received for your study in the ‘Funding Information’ section. 5. Thank you for stating the following financial disclosure: This work was developed under the FPU20/03664 doctoral contract granted by the Spanish Ministry of Universities. The authors acknowledge support from the PID2020-113316GB-100, PID2024-159020NB-100 and PID2024-159094NB-I00 research projects, funded by the Spanish Ministry of Science, Innovation and Universities; the COOPB23002 cooperation project of the CSIC, Spain; and the PNP project funded by CNRS/INSU. We also extend our gratitude to the ArchaeologyHub.CSIC research network for its support.    Please state what role the funders took in the study.  If the funders had no role, please state: "The funders had no role in study design, data collection and analysis, decision to publish, or preparation of the manuscript." If this statement is not correct you must amend it as needed. Please include this amended Role of Funder statement in your cover letter; we will change the online submission form on your behalf. 6. Thank you for uploading your study's underlying data set. Unfortunately, the repository you have noted in your Data Availability statement does not qualify as an acceptable data repository according to PLOS's standards. At this time, please upload the minimal data set necessary to replicate your study's findings to a stable, public repository (such as figshare or Dryad) and provide us with the relevant URLs, DOIs, or accession numbers that may be used to access these data. For a list of recommended repositories and additional information on PLOS standards for data deposition, please see https://journals.plos.org/plosone/s/recommended-repositories. 7. When completing the data availability statement of the submission form, you indicated that you will make your data available on acceptance. We strongly recommend all authors decide on a data sharing plan before acceptance, as the process can be lengthy and hold up publication timelines. Please note that, though access restrictions are acceptable now, your entire data will need to be made freely accessible if your manuscript is accepted for publication. This policy applies to all data except where public deposition would breach compliance with the protocol approved by your research ethics board. If you are unable to adhere to our open data policy, please kindly revise your statement to explain your reasoning and we will seek the editor's input on an exemption. Please be assured that, once you have provided your new statement, the assessment of your exemption will not hold up the peer review process. 8. We note that Figure 1 in your submission contain satellite images which may be copyrighted. All PLOS content is published under the Creative Commons Attribution License (CC BY 4.0), which means that the manuscript, images, and Supporting Information files will be freely available online, and any third party is permitted to access, download, copy, distribute, and use these materials in any way, even commercially, with proper attribution. For these reasons, we cannot publish previously copyrighted maps or satellite images created using proprietary data, such as Google software (Google Maps, Street View, and Earth). For more information, see our copyright guidelines: http://journals.plos.org/plosone/s/licenses-and-copyright. We require you to either present written permission from the copyright holder to publish these figures specifically under the CC BY 4.0 license, or remove the figures from your submission: a. You may seek permission from the original copyright holder of Figure 1 to publish the content specifically under the CC BY 4.0 license.   We recommend that you contact the original copyright holder with the Content Permission Form (http://journals.plos.org/plosone/s/file?id=7c09/content-permission-form.pdf) and the following text:“I request permission for the open-access journal PLOS ONE to publish XXX under the Creative Commons Attribution License (CCAL) CC BY 4.0 (http://creativecommons.org/licenses/by/4.0/). Please be aware that this license allows unrestricted use and distribution, even commercially, by third parties. Please reply and provide explicit written permission to publish XXX under a CC BY license and complete the attached form.” Please upload the completed Content Permission Form or other proof of granted permissions as an "Other" file with your submission. In the figure caption of the copyrighted figure, please include the following text: “Reprinted from [ref] under a CC BY license, with permission from [name of publisher], original copyright [original copyright year].” b. If you are unable to obtain permission from the original copyright holder to publish these figures under the CC BY 4.0 license or if the copyright holder’s requirements are incompatible with the CC BY 4.0 license, please either i) remove the figure or ii) supply a replacement figure that complies with the CC BY 4.0 license. Please check copyright information on all replacement figures and update the figure caption with source information. If applicable, please specify in the figure caption text when a figure is similar but not identical to the original image and is therefore for illustrative purposes only.The following resources for replacing copyrighted map figures may be helpful: USGS National Map Viewer (public domain): http://viewer.nationalmap.gov/viewer/The Gateway to Astronaut Photography of Earth (public domain): http://eol.jsc.nasa.gov/sseop/clickmap/Maps at the CIA (public domain): https://www.cia.gov/library/publications/the-world-factbook/index.html and https://www.cia.gov/library/publications/cia-maps-publications/index.htmlNASA Earth Observatory (public domain): http://earthobservatory.nasa.gov/Landsat:
http://landsat.visibleearth.nasa.gov/USGS EROS (Earth Resources Observatory and Science (EROS) Center) (public domain): http://eros.usgs.gov/#Natural Earth (public domain): http://www.naturalearthdata.com/ 9. We note that Figures 2 and SI1 in your submission contain copyrighted images. All PLOS content is published under the Creative Commons Attribution License (CC BY 4.0), which means that the manuscript, images, and Supporting Information files will be freely available online, and any third party is permitted to access, download, copy, distribute, and use these materials in any way, even commercially, with proper attribution. For more information, see our copyright guidelines: http://journals.plos.org/plosone/s/licenses-and-copyright. We require you to either present written permission from the copyright holder to publish these figures specifically under the CC BY 4.0 license, or remove the figures from your submission: a. You may seek permission from the original copyright holder of Figures 2 and SI1 to publish the content specifically under the CC BY 4.0 license.  We recommend that you contact the original copyright holder with the Content Permission Form (http://journals.plos.org/plosone/s/file?id=7c09/content-permission-form.pdf) and the following text:“I request permission for the open-access journal PLOS ONE to publish XXX under the Creative Commons Attribution License (CCAL) CC BY 4.0 (http://creativecommons.org/licenses/by/4.0/). Please be aware that this license allows unrestricted use and distribution, even commercially, by third parties. Please reply and provide explicit written permission to publish XXX under a CC BY license and complete the attached form.” Please upload the completed Content Permission Form or other proof of granted permissions as an "Other" file with your submission.  In the figure caption of the copyrighted figure, please include the following text: “Reprinted from [ref] under a CC BY license, with permission from [name of publisher], original copyright [original copyright year].” b. If you are unable to obtain permission from the original copyright holder to publish these figures under the CC BY 4.0 license or if the copyright holder’s requirements are incompatible with the CC BY 4.0 license, please either i) remove the figure or ii) supply a replacement figure that complies with the CC BY 4.0 license. Please check copyright information on all replacement figures and update the figure caption with source information. If applicable, please specify in the figure caption text when a figure is similar but not identical to the original image and is therefore for illustrative purposes only. 10. Please upload a new copy of Figures 1 – 6, as the detail is not clear. Please follow the link for more information:  https://journals.plos.org/plosone/s/figures 11. Please include captions for your Supporting Information files at the end of your manuscript, and update any in-text citations to match accordingly. Please see our Supporting Information guidelines for more information: http://journals.plos.org/plosone/s/supporting-information. 12. If the reviewer comments include a recommendation to cite specific previously published works, please review and evaluate these publications to determine whether they are relevant and should be cited. There is no requirement to cite these works unless the editor has indicated otherwise.

**Additional Editor Comments:**

Three independent reviews have now been received for this manuscript. All reviewers recognise the interest of the case study and the care taken in the field and laboratory work, but they also raise substantive concerns regarding the strength of the interpretations, the clarity of data presentation, and the alignment between the evidence and the conclusions drawn. In particular, the reviewers point to the need for a clearer separation between empirical results and interpretative hypotheses, as well as for improvements in structure, presentation and supporting documentation. In addition to the reviewers’ comments, I also outline below several points that I ask the authors to address in a revised version of the manuscript.

The CURRENT TITLE does not accurately reflect the evidential scope of the manuscript. It presents a specific interpretative explanation-intentional slag re-smelting in modern times-as an established result, whereas the study itself demonstrates only the occurrence of a late reheating event recorded archaeomagnetically in the upper parts of the furnaces. The causal mechanism responsible for this reheating is inferred and discussed, but not independently demonstrated, and alternative explanations are explicitly considered in the manuscript. The title therefore overstates the degree of interpretative certainty supported by the data. The authors are required to revise the title so that it reflects the documented observation and dating of a late reheating event, rather than asserting a specific technological practice as a confirmed outcome.

The current set of KEYWORDS largely duplicates terms already used in the title and therefore adds limited descriptive or indexing value. The keywords should be revised to avoid repetition and to better reflect the core problem addressed in the manuscript, rather than restating the general methodological or regional context.

The ABSTRACT overstates the study’s conclusions by presenting intentional slag re-smelting as a confirmed result, whereas the data demonstrate only a late reheating event recorded archaeomagnetically. The interpretative nature of the proposed mechanism and the existence of alternative explanations are not adequately reflected. The abstract must be revised to align its claims with the evidential limits of the study.

The overall STRUCTURE OF THE MANUSCRIPT requires revision. At present, empirical results are intermixed with interpretative elements, particularly in sections dealing with comparisons to geomagnetic field models and archaeomagnetic dating. These parts go beyond the presentation of measured data and belong analytically to interpretation rather than to the Results section.

The manuscript should be REORGANISED so that the Results section is limited to the presentation of empirical data and their statistical treatment, while archaeomagnetic dating and model-based comparisons are clearly separated and placed in an appropriate analytical or discussion section. In its current form, the structure obscures the boundary between results and interpretation and encourages overstatement of the study’s conclusions.

The CONCLUSIONS section goes beyond what is securely demonstrated by the data. While the archaeomagnetic results clearly indicate a late reheating event affecting the upper parts of the furnaces, the conclusions present intentional slag re-smelting as the most likely explanation, rather than as a working hypothesis. The limitations of sampling and the unresolved nature of the causal mechanism are not sufficiently reflected. The Conclusions should be revised to align strictly with the empirical findings, to state uncertainties explicitly, and to avoid privileging a single interpretative scenario where the evidence remains indirect.

ALL FIGURES use a grey background, which reduces clarity and is not appropriate for publication. The background should be removed and all figures should be prepared on a clean white background to improve readability and ensure consistency with journal standards.

The SUPPORTING INFORMATION requires substantial revision. At present, it contains several issues that undermine clarity and confidence in the presentation of the data. Large specimen-level tables include values that are difficult to interpret without additional guidance, including very large α95 confidence parameters and results based on extremely low numbers of specimens, which should either be explicitly justified or excluded from summary statistics. Correction factors for anisotropy and cooling rate vary widely, in some cases reaching unusually high values, yet these are not consistently explained or contextualised. Moreover, it is not always clear which data were retained for the final calculations and which are provided solely for completeness. The Supporting Information also mixes raw data, derived results and interpretative material without a clear structure. Finally, Figure SI1 is directly relevant to the interpretation of surface processes and sampling strategy and would be more appropriate in the main text. The Supporting Information should be carefully checked for numerical accuracy and internal consistency, streamlined, and reorganised so that it clearly supports the main manuscript rather than functioning as an unfiltered data dump.

FIGURE 1 requires revision in both its caption and cartographic design. The figure clearly consists of two panels, combining a regional location map of South America with a local-scale image of the Quillay site, yet the current caption treats it as a single satellite image and does not distinguish between these different spatial scales. In addition, the use of an orthophotomap as the main local representation is of limited analytical value. While visually informative, it does not adequately convey the geomorphological and topographic context that is relevant to the interpretation of surface exposure, site setting and potential post-depositional processes. A simplified topographic map, a DEM-based representation, or preferably a geomorphological map would better support the interpretative aims of the study. The authors should revise Figure 1 accordingly, either by replacing the orthophotomap or by complementing it with a more informative geomorphological or topographic representation.

TABLE 1. First, the table mixes dates obtained specifically for this study with dates taken from the literature, yet the implications of this heterogeneity are not discussed, and the referenced sources are not clearly cross-linked at the table level. Second, the treatment of multiple calibrated probability ranges (eg. for QLL09) is not sufficiently explained, and it is unclear how these alternative solutions are handled in the chronological interpretation. Third, the attribution of the QLL07 date to an old-wood effect is discussed in the text but not clearly flagged in the table itself, which may mislead readers treating the radiocarbon ages at face value. The table would benefit from clearer annotation indicating which dates are considered representative of kiln use and which are treated as problematic or interpretatively discounted.

TABLE 2. The rationale for treating uncorrected directions as final while applying anisotropy and cooling-rate corrections to intensity data is not explicitly explained in the table or its caption. In addition, the meaning of the nD/nD and nI/nI ratios is not immediately clear and should be defined more transparently. The table does not indicate whether any specimens were excluded from the final means or how specimen-level variability was propagated to kiln-level results. While Table 2 is acceptable in principle, these issues should be addressed to ensure full transparency and consistency with the methodological description and Supporting Information.

The presentation of TABLE 3 is problematic and requires revision. Although the table header indicates three global geomagnetic models, the structure of the table is ambiguous, as multiple age intervals are listed within individual cells without a clear explanation of their origin or status. It is not evident which intervals correspond to primary solutions and which represent alternative or secondary model outputs. In particular, early Common Era age ranges (e.g. 0-150 CE) are presented alongside late nineteenth–twentieth century ranges without clarification of their statistical relevance or interpretative weight.

Moreover, the table does not specify whether the reported age ranges are derived from directional data, intensity data, or their combination, nor does it indicate how multimodal probability solutions were treated. As a result, the table gives the impression that all age intervals are equally valid, despite their clear inconsistency with the archaeological context. This lack of hierarchy and explanation significantly reduces the clarity and interpretability of the results.

Table 3 should be restructured or expanded with explicit clarification of the modelling approach, the nature of multimodal solutions, and the criteria used to identify the age ranges discussed in the text. In its current form, the table does not meet the standards of transparent and unambiguous presentation of archaeomagnetic dating results.

Please also address several additional points that I consider potentially problematic, although I acknowledge that some of them may be open to interpretation and would benefit from clarification rather than outright correction:

In several places, archaeomagnetic dating is treated not only as a method for constraining the timing of a reheating event, but also as evidence for its causal mechanism. This distinction should be made clearer, as archaeomagnetic data date heating but do not, by themselves, identify whether the process was anthropogenic or natural.

The pronounced mismatch between the radiocarbon chronology and the archaeomagnetic ages is described but not sufficiently analysed as a methodological and interpretative issue. A more explicit discussion is needed to explain why the radiocarbon dates consistently align with the archaeological context while archaeomagnetism records a much later episode.

The sampling strategy is limited to the upper parts of the furnaces, yet conclusions are drawn about the reuse and function of the structures as a whole. This limitation should be more clearly reflected in the scope and strength of the interpretative claims.

alternative natural heating processes, such as surface fires, are mentioned but not explored in a fully balanced way. It should be clarified which aspects of the archaeomagnetic signal cannot be readily explained by natural processes and which remain ambiguous.

Intensities are reported after anisotropy and cooling-rate corrections, whereas directional data are treated as final without equivalent correction. Even if methodologically justified, the rationale for this approach should be stated more clearly to ensure internal consistency.

Throughout the manuscript, the distinction between empirical results, interpretative inferences, and working hypotheses is not always sufficiently clear, particularly with respect to the proposed reuse scenario, which at times appears more certain than the available evidence allows.

Reviewers' comments:

Reviewer's Responses to Questions

**Comments to the Author**

1. Is the manuscript technically sound, and do the data support the conclusions?

Reviewer #1: Partly

Reviewer #2: Yes

Reviewer #3: Partly

2. Has the statistical analysis been performed appropriately and rigorously? 

Reviewer #1: I Don't Know

Reviewer #2: Yes

Reviewer #3: Yes

3. Have the authors made all data underlying the findings in their manuscript fully available?

Reviewer #1: Yes

Reviewer #2: No

Reviewer #3: Yes

4. Is the manuscript presented in an intelligible fashion and written in standard English?

Reviewer #1: Yes

Reviewer #2: Yes

Reviewer #3: No

5. Review Comments to the Author

**Reviewer #1:** This is a very interesting paper. I know some of the authors personally and by reputation and they are an excellent team.

Some thoughts:

The authors argue that kilns on a prehistoric Inka site were reused in the modern period during the 19th or 20th century.

The Inka dating of Quillay is definitively demonstrated by artifacts and carbon dates. The archaeological data indicate that the kilns were used for copper production in the pre-Columbian period.

The modern (19th and 20th c.) reuse of the kilns is suggested by archaeomagnetic dating. However, there are no associated 19th or 20th century artifacts or carbon dates. No historical data to support the reuse are presented.

There are two hypotheses presented: 1) the site was reused (Reuse hypothesis) in the modern period or 2) the data are the result of wildfires (Wildfire hypothesis). The title of the paper betrays the authors’ preferred hypothesis--that this is a modern reuse of the ancient technology. However, the authors also clearly state that that the archaeomagnetic data could have been produced from wildfires.

It seems more effective to present both hypotheses as feasible upfront and then outline the data that supports/rejects each of these. It might be a better strategy than to put the Wildfire hypothesis in the Discussion.

Data supporting the kiln reuse hypothesis are outlined in the paper, but the wildfire hypothesis is not presented as forcefully. It strikes me that both explanations are viable. Given what is presented in the manuscript. I would say that the wildfire hypothesis is more strongly supported that the modern reuse one. I therefore strongly recommend changing the title to reflect two viable possibilities. I would take out “slag re-smelting in modern times.”

The authors state that this kiln reuse is a “practice documented at other Andean metallurgical sites”. Likewise, they state that “historical records of metal recycling in the region during the late 19th and early 20th centuries documented in other Andean metallurgical sites” supports their favored hypothesis. This would be strong evidence for the Reuse hypothesis. However, I can find no description of such sites in their literature review.

Therefore, they must include brief descriptions and references to these historical records. The manuscript mentions: “small-scale metallurgical operations carried out by marginal or impoverished individual workers (56,61,62)”. The titles of the two of the three references do not seem to cover the modern period. Reference 62 lists “1st to 15th centuries”. I did a brief read of this outstanding article, but I could not find a reference to the modern reuse of pre-Columbian sites. I may have missed something. Ref. 56 is about fire management. Perhaps a relevant quote from ref. 56 would be useful. I could not access the manuscript. It should be reproduced in Supplementary materials. Ref 61 seems appropriate but some brief description of the sites would be warranted. I did a quick read and could not find any references to the reuse of kilns from pre-Colonial archaeological sites, though I could have missed something. The manuscript mentions Lechtman in Ref. 59 but the article is by another author (should this be ref. 60?). The provided link to this article did not open but I was able to get it directly. Again, my superficial read did not find any reference to the reuse of ancient sites, but I may have missed it.

The article by Lechtman (Ref. 60) does list several metallurgy sites with modern, Colonial and pre-Colonial occupations. These could be excellent candidates to support the modern reuse hypothesis. To be convincing, the authors should discuss two or more of these sites showing how Quillay fits into a broader pattern. This would strongly support the Reuse hypothesis and Quillay.

A major observation against the Reuse hypothesis is: why are there no 19th or early 20th century artifacts at Quillay similar to those that dated the site as Inka? One would expect at least a few artifacts from the modern period if people worked the kilns. The photographs in ref. 57 of the Pirquineros shows plenty of material artifacts and other refuse surrounding their work sites. This absence must be explained. Also, why are there no carbon dates that came back as modern? Perhaps these were not sampled. If so, simply state this.

Finally, the Reuse hypothesis would be supported if there was an historical settlement nearby. GoogleEarth shows no settlements and the manuscript is silent on this issue.

Additional observations:

Dates should be expressed in calibrated formats, e.g. 1400 calAD or calCE.

Ln 85: “infantile” should be “infant”.

Ln 133: “Samples marked with an asterisk were dated specifically for this study at the CNA; the rest were retrieved from the available literature.” The “rest” of these dates must be specifically cited with full reporting format.

Table 1: Carbon dates should include the δ13c along with the nature of the material sampled--wood, seeds, carbon flecks etc. Some labs do not report δ13c in which case you simply write “N/A”.

The date for kiln QLL07 is dismissed as too old. However, the 14c date provided overlaps (barely to be sure) the 2-sigma of three other dates and cannot be considered as an outlier. The old wood factor is certainly a possibility. In this case, it would be good to know what the material was that they dated. The δ13c could clarify this.

Ln. 147: “coherent’ should be “consistent”

Ln. 191 “Our data selection approach is consistent with recent high-quality archaeointensity and archaeodirectional studies (e.g. 38,39).” Refs. 38 and 39 include the corresponding author of the present manuscript as a co-author. It would be more effective to cite an independent international standard or another independent lab.

Fig. SI1 is very important and should be in the main text in my opinion.

To summarize:

Data to support the Reuse hypothesis include:

Confirmation that there was an earlier reuse. In this case it is the Inca and this is confirmed

Historical data showing that reuse occurred in other similar contexts. None is presented.

Archaeological remains from the time of the supposed reuse. None is presented.

Archaeometric (14c or archaeomagtic). No 14c but archaeomag is presented.

Settlement data demonstrating that there was an early Modern period settlement. No data are presented.

Data to support the wildfire hypothesis:

Historical data showing that wildfires occurred. Historical data are presented.

Lack of archaeological data since wildfires are natural. Confirmed.

Lack of a Modern settlement on or near the kilns. Unclear.

In short, assuming that the field and lab methods were not faulty, the Wildfire hypothesis is supported, the Reuse hypothesis is not.

**Reviewer #2:** The authors report on the interesting case when intensity and direction of the Earth’s magnetic field recovered from archaeological material are strongly discordant from radiocarbon dates. At the same time, both archaeomagnetic and radiocarbon data appear quite reliable. Therefore, this discordance between the two is real and needs an explanation. The authors propose that Inca XV century copper-smelting kilns were re-used at a later time, apparently in the end of XIX to beginning of XX century, to recover residual metal from slags. While it is desirable to further confirm this hypothesis by archaeological means and/or historical records, the study itself is of high quality and merits publication in PLoS One, after perhaps a minor revision along the lines specified below.

1) Experimental setup.

It would be helpful if the authors include the detailed description of experimental setup in the Supporting information, in addition to referring a reader to previous studies.

In the description of Thellier experiments, list the normal cooling time. How exactly was the CR correction calculated? Did you acquire a total TRM by cooling for 24 hours and then compare it with TRM acquired during a ‘normal’ cooling?

For the anisotropy correction, list temperature(s) at which the additional pTRMs in ±X, Y, Z directions were acquired for each specimen. This information can be conveniently included in the Table SI1.

For hysteresis measurements, detail how the data were corrected for the high-field slope. Note that the samples like the one shown in Fig. 3d do not reach magnetic saturation in the 1 T field, and therefore the Ms value would be underestimated, perhaps significantly. Please check the field used to measure thermomagnetic curves (lines 199-200). 30 mT seems to be a typo; curves shown in Fig. 3 look like as if they were measured in a near-saturation field.

2) Rock magnetism.

Samples used in this study are in most cases magnetically soft, with the exception of those like Q09.02r. At the same time, it seems an overly strong statement to refer to the coercive force ranging from 13 mT to 44 mT as “very low” (line 228). For a magnetite/maghemite-like phase, ‘very low’ would be something like <3-4 mT. Note that such samples would be dominated by multidomain grains and thus unsuitable for Thellier paleointensity experiments.

Considering the magnetic mineralogy, a conventional interpretation of a magnetically soft phase as low-Ti substituted or even pure magnetite (line 238) is probably not entirely correct. In archaeological ceramics sensu lato, magnetic phase(s) in most cases form during baking at high temperature, rather than inherited from a parent material. Ti is not the most likely element to substitute iron in the magnetite lattice during such a process; Al is equally if not more possible. In addition, vacancies could arise as well forming a phase intermediate between magnetite and maghemite [cf. Pilipenko, O. V., et al. (2019), Archaeomagnetic studies of the material of the archaeological monument Dmitrievskaya Sloboda II of the second millennium B.C, in Recent Advances in Rock Magnetism, Environmental Magnetism and Paleomagnetism, D. Nurgaliev, V. Shcherbakov, A. Kosterov and S. Spassov (Eds.), pp. 97-107, Springer International Publishing, Cham]. On the other hand, (near)-stoichiometric magnetite is not typical for ceramics, as attested by absence, in most cases, of the low-temperature Verwey phase transition [Kosterov, A., et al. (2021), High-coercivity magnetic minerals in archaeological baked clay and bricks, Geophys. J. Int., 224(2), 1256-1271; Troyano, M., et al. (2021), Analyzing the geomagnetic axial dipole field moment over the historical period from new archeointensity results at Bukhara (Uzbekistan, Central Asia), Phys. Earth Planet. Inter., 310, 106633].

What do the authors mean under “a reversible peak between 100 and 200ºC (Figure 3c)” (lines 249-250)? I rather see a Ms(T) curve that is slightly irreversible below 200°C. Note that a similar but even stronger irreversibility is observed for the specimen shown in Figure 3d, that apparently contain an ε-Fe2O3-like phase in the initial state. During the Ms(T) run, extra epsilon-phase apparently forms, as attested by cooling curve going noticeably higher than the heating one. Similar but less pronounced process could explain the irreversibility of Ms(T) curve for the specimen Q22.03r (Figure 3c) as well.

It is mentioned (lines 201-202) that susceptibility vs. temperature curves were measured for selected samples. Please show a few examples in the SI section.

Finally, it would be helpful if hysteresis parameters and Curie temperatures for all measured specimens were presented as a table in the Supporting Information. Include also the specimens’ group according to color classification.

3) Archaeomagnetic directions and intensities.

Authors accept as final and use for dating the directional results uncorrected for anisotropy whereas the intensities are corrected. This seems an inconsistent approach. It would be more straightforward to carry out dating using both uncorrected and corrected direction/intensity data sets.

Inspection of archaeointensity at the specimen level (Table SI1) shows that there are samples that require very large anisotropy correction, e.g. 07903B1 from kiln QLL07 and 9191 from QLL09. Resulting ‘corrected’ intensities deviate significantly from site means and are therefore suspect. Perhaps, it would be more prudent to reject intensities obtained from specimens requiring more than 10 % anisotropy correction.

In Table SI2, correction factors should be listed as well.

In Table SI3, entries showing final results for kiln QLL09 apparently contain multiple typos. Values for corrected declination and inclination appear interchanged, and α95s seem wrong.

4) Archaeomagnetic dating.

The comparison of archaeomagnetic directions and intensities is only made to global geomagnetic models (Figure 5). The problem is that these models use very few data from South America as an input. Perhaps it would be instructive to compare the intensity data obtained in this study also with the most recent South American reference curve [Goguitchaichvili, A., et al. (2023), Evolution of the Earth's magnetic field strength in northwestern Argentina during the last two millennia: Towards the improvement of south American geomagnetic paleosecular variation curve, J. South Amer. Earth Sci., 126, 104357].

5) Paper structure and presentation.

Text in lines 320 to 395 would be more appropriately placed in a separate section which may be provisionally entitled ‘Archaeomagnetic dating’. The Discussion section would then start at line 397.

Some figures are of inferior quality. In particular, Figure 3 is almost illegible. Please consider using different color scheme(s) to improve legibility.

Andrei Kosterov

December 29, 2025

**Reviewer #3:** In my view, the work presented in this manuscript remains preliminary and incomplete. Throughout the manuscript, I found myself wondering why the authors sampled only the upper part of the combustion structure. The authors themselves acknowledge this limitation (line 451 and following). However, sampling the combustion chamber and the combustion floor should not be considered “beyond the scope of this paper,” as stated in line 465. Such sampling would have allowed a more informed discussion of the two hypotheses proposed to explain the archaeomagnetic results.

In addition, a brief discussion of the radiocarbon (¹⁴C) ages appears necessary. Why do these ages agree with the archaeological chronology, and why do they differ so markedly from the archaeomagnetic age? Furthermore, could the presence of significant amounts of magnetic metals near the furnaces have locally distorted the Earth’s magnetic field? To my knowledge, no experimental archaeomagnetic study has demonstrated the feasibility of reliable archaeomagnetic dating of metallurgical furnaces processing iron ore. Although the furnace studied here was used for copper extraction, it remains unclear whether iron ore was entirely absent.

A complete sampling of the combustion structures would have reduced these uncertainties, including those related to the possible metallurgical function of the furnaces (particularly if the archaeomagnetic age had coincided with the ¹⁴C age). In archaeological contexts, it is relatively common for both ¹⁴C and archaeomagnetic ages to be younger than the archaeological age due to later reuse of structures. However, large discrepancies between ¹⁴C and archaeomagnetic ages are uncommon. This discrepancy is what makes the present study potentially unique, and it is unfortunate that the underlying cause cannot be more precisely identified.

There is no doubt that the work was carried out carefully and rigorously. However, my reading was repeatedly interrupted by inaccuracies in terminology, typographical errors, and imprecise formulations.

Below, I list a number of specific points, in order of appearance in the manuscript, that I believe should be addressed prior to publication.

1/ Title and throughout the manuscript: The manuscript uses the spelling “Inka” when referring to the Inca civilization. While “Inca” is the conventional spelling in English academic literature, “Inka” is sometimes used to reflect Quechua-based transliterations or specific cultural perspectives. If the use of “Inka” is intentional, this choice should be briefly justified and applied consistently throughout the manuscript. Otherwise, for clarity and consistency, the use of “Inca” is recommended.

2/ Title : The title may need to be revised, as this hypothesis is not clearly supported by the results presented in this study.

3/ Line 156: The orientation procedure of the plaster caps is unclear. Were sun sightings performed, or was only the magnetic azimuth measured? In the latter case, it is essential to explain how the magnetic declination correction was applied. In archaeomagnetism, high precision in sample orientation is critical, as even small orientation errors can result in significant temporal uncertainties when remanent magnetization directions are compared with secular variation reference curves.

4/ Line 160: Standard archaeomagnetic/paleomagnetic samples typically have a volume of approximately 8 cm³, not 2 cm³. Is this a typographical error, or were mini-samples used?

5/ Line 182 and Supporting Information table: Reference #36 is incorrect, as it refers to a second paper by Gómez-Paccard (PEPI, 2006), which should be cited instead. The cooling-rate correction is grounded in physical principles (e.g., Néel theory) and is justified by the fact that, for single-domain grains, thermoremanent magnetization increases with slower cooling rates. Consequently, this correction necessarily reduces the estimated archaeointensity. When F_cr > 1.0, the correction should therefore not be applied. Although this error has appeared in several published studies, it should not be perpetuated unless theoretical justification is provided showing that cooling-rate correction can increase archaeointensity estimates. In archaeomagnetism, the formulation proposed by Veitch (1984) has often been used; this formulation differs from that adopted here, as Veitch divides the intensity by the correction factor rather than multiplying it. Additional details and clarification regarding the cooling-rate correction are therefore required.

6/ Line 295: The sentence “Directional means were calculated using Fisher statistics” is awkward and potentially misleading. The mean direction obtained from the vector sum is independent of any distributional assumption, whereas the Fisher distribution applies only to the estimation of confidence parameters (e.g., κ, α95). For clarity and statistical rigor, these two steps should be explicitly distinguished (e.g., “the mean direction was calculated using the vector sum, and statistical parameters were computed assuming a Fisher distribution”), as stated in the reference cited by the authors.

7/ Line 298: It is unclear how anisotropy correction could modify the MAD. In general, anisotropy correction is applied to the ChRM direction and does not affect the MAD. Even if the correction were applied to individual vectors during demagnetization, it is difficult to see how the MAD could change. This point requires clarification.

8/ Line 306: Please add confidence cones.

9/ Line 312: It is not the mean directions themselves that overlap, but rather their 95% confidence cones. Statistical tests would be required to support this interpretation.

10/ Line 313 (Figure 5a): This representation is not a stereographic projection. The plotting method is unconventional and potentially misleading. In equal-area or equal-angle stereographic projections, the y-axis does not correspond to declination. For example, a direction with I = 0 and D = 45 would lie outside the diagram shown. In addition, by convention in archaeomagnetism/paleomagnetism, negative inclinations are represented by open symbols rather than filled symbols.

Line 322: Replace “circles of confidence” with “confidence cones.”

6. PLOS authors have the option to publish the peer review history of their article (what does this mean?). If published, this will include your full peer review and any attached files.

Reviewer #1: No

Reviewer #2: **Yes:** Andrei Kosterov

Reviewer #3: No

---

## [Author Response · Author response to Decision Letter 1]

26 Mar 2026

Please, you can find a detailed response of the comments in the attached pdf

---

## [Editor Report · Decision Letter 1]

20 Apr 2026

PONE-D-25-64919R1Archaeomagnetic evidence shows Inka metallurgical kilns were affected by a contemporary re-heating event (Quillay, NW Argentina)PLOS One

Dear Dr. Gómez-Paccard,

Thank you for submitting your manuscript to PLOS ONE. After careful consideration, we feel that it has merit but does not fully meet PLOS ONE’s publication criteria as it currently stands. Therefore, we invite you to submit a revised version of the manuscript that addresses the points raised during the review process.<section class="text-token-text-primary w-full focus:outline-none [--shadow-height:45px] has-data-writing-block:pointer-events-none has-data-writing-block:-mt-(--shadow-height) has-data-writing-block:pt-(--shadow-height) [&:has([data-writing-block])>*]:pointer-events-auto [content-visibility:auto] supports-[content-visibility:auto]:[contain-intrinsic-size:auto_100lvh] R6Vx5W_threadScrollVars scroll-mb-[calc(var(--scroll-root-safe-area-inset-bottom,0px)+var(--thread-response-height))] scroll-mt-[calc(var(--header-height)+min(200px,max(70px,20svh)))]" data-scroll-anchor="false" data-testid="conversation-turn-36" data-turn="assistant" data-turn-id="request-WEB:8d92092a-f634-416b-8192-2534a966b1ed-18" dir="auto">

The manuscript is potentially suitable for publication in PLOS ONE, but several issues affecting clarity, transparency, and evidential precision still need to be resolved before it can be accepted. In particular, the title, abstract, keywords, and presentation of the chronological results should be revised to reflect the data more accurately, and the argument currently supported by personal communication should either be substantiated with citable evidence or clearly reduced in interpretative weight. The required revisions are outlined below; any wording suggestions are offered to facilitate revision, but the essential requirement is that the revised manuscript presents its results and interpretations with sufficient precision and transparency to meet PLOS ONE’s publication criteria.

</section>Please submit your revised manuscript by Jun 04 2026 11:59PM. If you will need more time than this to complete your revisions, please reply to this message or contact the journal office at plosone@plos.org. Please include the following items when submitting your revised manuscript:

We look forward to receiving your revised manuscript.

Kind regards,

Przemysław Mroczek, Dr. hab.

Academic Editor

PLOS One

Journal Requirements:

Additional Editor Comments:

Thank you for submitting the revised version of the manuscript. The paper has improved substantially. In particular, the title and abstract are now better aligned with the evidential scope of the study, the manuscript is more clearly structured, and the Discussion reflects the limitations of the dataset in a more balanced way. I therefore consider the manuscript suitable for publication in principle, but I would still ask for a minor revision before it can be accepted.

The remaining issues are limited in number, but they should be addressed carefully and explicitly in a point-by-point response.

First, the title, abstract, and keywords should be revised further for clarity and precision. The current title is much improved, but it remains somewhat heavy stylistically, and terms such as “shows” and “contemporary” still read as slightly stronger or less precise than necessary. A cleaner wording such as Archaeomagnetic evidence indicates post-Inka reheating of metallurgical kilns at Quillay (NW Argentina) may better reflect the evidential scope of the study. The abstract is also improved, but several formulations still seem somewhat overstated in relation to the dataset, particularly “well-dated”, “strongly deviate”, and “clearly reveal”. The wording of the archaeomagnetic age estimate should also be aligned more closely with the date ranges discussed in the main text. One possible revised abstract could read as follows:

This study presents an archaeomagnetic analysis of five metallurgical kilns from the Inka settlement of Quillay (Catamarca, NW Argentina), previously attributed to the Inka period on the basis of radiocarbon dates and archaeological evidence. All five structures yielded consistent palaeofield values, and both directional and intensity determinations are statistically indistinguishable, suggesting that the sampled parts of the kilns were reheated within a relatively short interval. Comparison with geomagnetic field models, particularly BIGMUDI4k.1, indicates that this last heating event occurred between the late nineteenth and mid-twentieth centuries, rather than during the Inka occupation. The discrepancy between the archaeomagnetic and radiocarbon evidence is therefore consistent with a later remagnetisation event affecting structures originally used in Inka times. Possible explanations include a local wildfire or the deliberate re-use of the upper chambers to recover copper from slag. The results refine the occupational history of Quillay and support the interpretation of a later thermal event affecting structures originally used in Inka times.

Likewise, the keywords are generally relevant, but they should be made more specific and more consistent with the revised focus of the manuscript. One possible set would be: archaeomagnetism; archaeomagnetic dating; archaeometallurgy; Andean archaeology; kiln reheating.

Second, please improve the presentation of Table 3. The table is informative, but its chronological message is not yet fully self-explanatory, particularly where some models yield disjoint age intervals within the 95% confidence range. Please clarify this more explicitly in the table layout, caption, or accompanying text, so that the reader can understand the meaning of these alternative intervals without having to infer the interpretation from the discussion alone..

Third, please revise the evidential status of the argument supporting the re-smelting hypothesis. Part of this discussion currently relies on a statement attributed to “Pablo Cruz, personal communication”. This point should not remain in its current form. It should either be supported by citable and accessible evidence or reduced to a clearly marked ancillary observation that does not carry substantial interpretative weight.

Finally, the manuscript requires a careful final language and copy-editing pass. There are still scattered typographical and editorial issues in the text, including missing spaces, minor spelling problems, and small punctuation inconsistencies. Please also ensure that the same cautious wording is maintained consistently across the title, abstract, Discussion, Conclusions, and figure captions.

These are, in my view, minor but necessary revisions. Once they are addressed satisfactorily, I see no major obstacle to publication.

---

## [Author Response · Author response to Decision Letter 2]

7 May 2026

Thanks for all your suggestion, which seems appropriate to all authors and have been

accepted. In the attached pdf you can find a point-by-pint answer to all the issues you raised.

---

## [Editor Report · Decision Letter 2]

10 May 2026

Archaeomagnetic evidence indicates post-Inka reheating of metallurgical kilns at Quillay (NW Argentina)

PONE-D-25-64919R2

Dear Dr. Gómez-Paccard,

We’re pleased to inform you that your manuscript has been judged scientifically suitable for publication and will be formally accepted for publication once it meets all outstanding technical requirements.

Kind regards,

Przemysław Mroczek, Dr. hab.

Academic Editor

PLOS One

Additional Editor Comments (optional):

Thank you for submitting the revised version of the manuscript and for providing a clear point-by-point response to the remaining editorial comments.

I am satisfied that the main issues raised in the previous decision have been addressed. The title, abstract and keywords are now more precise and better aligned with the evidential scope of the study. The presentation and explanation of Table 3 have also been improved, and the status of the re-smelting hypothesis is now more appropriately framed, without relying on personal communication as a substantive element of the argument. I also appreciate the more cautious wording used in the Discussion, Conclusions and limitation section.

Only minor typographical and copy-editing issues appear to remain, and these can be handled during the final editorial and production stages. In my view, the manuscript now meets the requirements for publication in PLOS ONE.

Thank you again for your careful revisions.
---

## [Editor Report · Acceptance letter]

PONE-D-25-64919R2

PLOS One

Dear Dr. Gómez-Paccard,

I'm pleased to inform you that your manuscript has been deemed suitable for publication in PLOS One. Congratulations! Your manuscript is now being handed over to our production team.

Kind regards,

on behalf of

Dr. hab. Przemysław Mroczek

Academic Editor

PLOS One